# Prevalence and Epidemiology of Multidrug-Resistant Pathogens in the Food Chain and the Urban Environment in Northwestern Germany

**DOI:** 10.3390/antibiotics9100708

**Published:** 2020-10-16

**Authors:** Sylvia Klees, Natalie Effelsberg, Birgit Stührenberg, Alexander Mellmann, Stefan Schwarz, Robin Köck

**Affiliations:** 1Department of Microbiology and Zoonoses, Chemisches und Veterinäruntersuchungsamt Ostwestfalen-Lippe, 32758 Detmold, Germany; Birgit.Stuehrenberg@cvua-owl.de; 2Institute of Hygiene, University Hospital Münster, 48149 Münster, Germany; Natalie.Effelsberg@ukmuenster.de (N.E.); mellmann@uni-muenster.de (A.M.); r.koeck@drk-kliniken-berlin.de (R.K.); 3Department of Veterinary Medicine, Institute of Microbiology and Epizootics, Freie Universität Berlin, 14163 Berlin, Germany; stefan.schwarz@fu-berlin.de; 4Institute of Hygiene, DRK Kliniken Berlin, 14050 Berlin, Germany

**Keywords:** antimicrobial resistance, food safety, one health, zoonosis

## Abstract

The surveillance of antimicrobial resistance among humans and food-producing animals is important to monitor the zoonotic transmission of multidrug-resistant bacteria (MDRB). We assessed the prevalence of four MDRB within the meat production chain, including extended-spectrum β-lactamase (ESBL)-producing, carbapenemase-producing Enterobacterales (CPE) and colistin-resistant Enterobacterales (Col-E), as well as vancomycin-resistant enterococci (VRE). In total, 505 samples from four stages of meat production, i.e., slaughterhouses, meat-processing plants, fresh food products and the urban environment, were collected in northwestern Germany in 2018/2019 and screened for the presence of MDRB using both culture-based and PCR-based techniques. We detected genes encoding for carbapenemases in 9–56% (*bla*_OXA-48_, *bla*_KPC_, *bla*_NDM_*, bla*_VIM_) and colistin resistance-encoding *mcr* genes in 9–26% of the samples from all stages. Culture-based analysis found CPE and VRE only in environmental samples (11% and 7%, respectively), but Col-E and ESBL-producers in 1–7% and 12–46% of samples from all stages, respectively. Overall, our results showed that ESBL-producers and *mcr*-carrying Col-E were common in food-producing animals at slaughterhouses, in meat-processing plants and in food items at retail, while CPE and VRE were only found in the environment. The discrepancy between detected carbapenemase genes and isolated CPE emphasizes the need for more sensitive detection methods for CPE monitoring.

## 1. Introduction

The growing prevalence of multidrug-resistant bacteria (MDRB), which are commonly understood as bacteria with a non-susceptibility to three or more classes of antibiotics [1], is one of the major global threats for human and animal healthcare and wellbeing. It is known that MDRB from animals can be transferred to humans (or vice versa) either by (in)direct contact or by the oral ingestion of contaminated, animal-based food items [2]. Therefore, the surveillance of antimicrobial resistance among bacteria from humans and food-producing animals as well as the analysis of molecular typing data to monitor their transmission are important to understand and combat the spread of MDRB.

To address this issue, programs for the monitoring of MDRB in food-producing animals have been established in the European Union (EU). These programs are based on the detection of (i) foodborne pathogens (e.g., *Salmonella* spp.), which are subjected to antimicrobial susceptibility testing (AST); (ii) defined MDRB (e.g., methicillin-resistant *Staphylococcus aureus*, MRSA) using selective culture-based “screening” techniques to reliably identify these MDRB, and (iii) commensal *Escherichia coli* from food-production environments to obtain a global impression on antimicrobial susceptibility among these isolates.

In recent years, the occurrence of several types of MDRB was highlighted both in individual studies and in EU zoonoses-monitoring programs. Besides MRSA, these mainly included extended-spectrum β-lactamase- (ESBL) or AmpC-producing Enterobacterales, carbapenemase-producing Enterobacterales (CPE), and colistin-resistant Enterobacterales (Col-E). In the 1990s, the occurrence of vancomycin-resistant enterococci (VRE) in livestock was also addressed. In Germany, previous studies showed that MRSA affected 40–100% of all pig and chicken farms [3,4,5,6]. Moreover, ESBL-*E. coli* were found in 61–100% of these farms [3,4]. CPE were only occasionally detected [7], while Col-E were found in 26% of the pig farms in 2011/2012 [8].

In this study, we used the EU-wide monitoring program of 2018/2019, which was in concordance with the EU directive on the monitoring of zoonoses and zoonotic agents (2003/99/EC) and decision 2013/652/EU, to detect MDRB in the meat production chain. We extended this analysis by adding samples from meat-processing plants and the urban environment and performing additional genotypic and phenotypic investigations to overcome some limitations of the regular monitoring.

CPE-monitoring in EU-programs is based on culture-based techniques. However, some studies indicate that the culture-based detection of CPE is challenging in environmental samples due to overgrowth (e.g., by *Pseudomonas* spp.) or by the low-level expression of the carbapenemase-encoding genes as reviewed by Köck et al. [7]). On the other hand, carbapenemase-encoding genes detected using polymerase chain reaction (PCR) can also originate from environmental species such as Shewanella spp. [9]. Therefore, a combined approach including both techniques should improve CPE detection.

While Col-E are occasionally detected when screening for commensal E. coli, no selective screening for Col-E is applied within the EU-wide monitoring program. Recently, it was reported that Col-E harboring the plasmid-located colistin resistance-mediating *mcr* genes had emerged in livestock [10,11]. Here, we aimed to assess the prevalence of Col-E along the meat production chain, to characterize the *mcr* genes, and to determine whether very recent reductions of colistin use in Germany already led to a decrease in Col-E. As culture-based detection alone was reported to be difficult due to overgrowth by intrinsically resistant species [12], it seems feasible to use a combined approach to selectively screen for Col-E and *mcr* genes.

Although the transfer of VRE from livestock-related sources to humans is considered to play a minor role in the epidemiology of VRE among humans, we aimed to assess the culture-based prevalence of VRE in the meat production chain due to a recent increase in VRE in human healthcare in Germany [13].

The aim of this study was to determine the current state of MDRB prevalence in the food chain in Germany using a systematic sampling strategy covering different stages of meat production and to enhance the sensitivity of screening methods by combining PCR and culture-based approaches (for CPE and Col-E).

## 2. Results

### 2.1. Prevalence of AMR Genes

In total, 505 samples from the different stages of the meat production chain and the urban environment were screened for the presence of ESBL/AmpC-producing Enterobacterales, Col-E, CPE, and VRE (Figure 1). The samples were taken from slaughterhouses, meat-processing plants, fresh food products and environmental sources as illustrated in the methods section. The presence of both CPE and Col-E was investigated using PCR as well as culturing. For both pathogens, we found substantially more positive samples in the PCR than in the culture-based approach.

The prevalence of MDRB varied between the different stages of meat production and the animal species (Table 1). In our PCR screening for CPE, we detected *bla*_OXA-48_ in samples of all stages of the meat production chain; specifically *bla*_OXA-48_ was common in samples from meat plants and on carcasses of broilers and pigs but not in their cecum content. In contrast, *bla*_VIM_, *bla*_NDM,_ and *bla*_KPC_ were only detected in environmental samples from sewage and sludge. Using the culture-based approach, CPE isolates were only successfully cultured from environmental samples and included *Citrobacter freundii* (*n* = 2), *E. coli* (*n* = 2) and *Klebsiella oxytoca* (*n* = 2) carrying *bla*_OXA-48_ and *Klebsiella pneumoniae* (*n* = 2) carrying *bla*_KPC._

Genotypically, *mcr* genes were detected in 18% of all 505 samples. From these samples, 21 Col-E isolates were successfully cultured. Most of them were *mcr-1*-positive *E. coli* (*n* = 18). Additionally, we found *mcr-1*-positive *K. pneumoniae* (*n* = 1) and *Morganella morganii* (*n* = 1)*,* and another *M. morganii* isolate, which carried *mcr-5* (*n* = 1).

Using only the culture-based approach, we detected VRE (*Enterococcus faecalis*) in a single sample from the neck skin of a broiler in a slaughterhouse, and in four environmental samples from sewage (all *E. faecium*). All other samples were negative for VRE.

We were able to culture ESBL/AmpC-producing Enterobacterales from samples originating from all the stages of meat production and from sewage, sludge, and soil, but not from water. Poultry meat was more often contaminated with ESBL/AmpC-producing Enterobacterales than pork or beef. Most isolates were identified as *E. coli* (*n* = 79), two as *K. pneumoniae,* and one as *Enterobacter* spp.

### 2.2. Genotyping of Selected Isolates

A subset of CPE, Col-E, and VRE isolates was chosen for whole genome sequencing to determine their molecular characteristics, particularly resistance genes, and their clonal relationship using core genome multilocus sequence types (cgMLST).

Among the CPE isolates, two *E. coli* strains carrying *bla*_OXA-48,_ which were isolated from sewage and sludge, respectively, differed in only one cgMLST allele, indicating that they shared the same origin. For the other isolates, no clonal relationship was detected.

None of the ten *mcr-*positive isolates that were sequenced shared the same clonal background. However, the circular *mcr*-1-positive contigs, most likely resembling plasmids, belonged to the same two types of replicon families, IncX4 (*n* = 6) and IncHI2 (*n* = 2). The *mcr*-5 gene carried by a *M. morganii* isolate was located on a novel, unique 51,340 kb plasmid, which generated no hits in the NCBI nucleotide database, covering more than 52% of its sequence. Notably, no other known gene conferring antimicrobial resistance was found on this plasmid.

Among the VRE isolates, three of the four sequenced *E*. *faecium* isolates carried *vanB* and belonged to the MLST sequence type (ST) 117 and the remaining was ST80 and carried *vanA.*

The selected isolates were also screened for the presence of known resistance genes and characterized by phenotypic AST. The results are shown in Figure 2, Figure 3 and Figure 4.

## 3. Discussion

The aim of this study was to assess the prevalence of MDRB in the food chain in Germany by extending the existing EU-wide monitoring program with additional samples and improving the sensitivity of screening methods by combining PCR and culture-based approaches.

This was particularly important for the detection of carbapenemase genes because difficulties to find CPE in culture-based analyses from livestock had been reported. The CPE found (specifically the CPEs encoded by *bla*_OXA-48_-like genes) often exhibited low MICs, which might lead to false-negative culture results [14]. Therefore, we tried to facilitate the identification of relevant samples by combining culture-based with genotypic screening. We found notably more carbapenemase-encoding genes in the PCR screening than we isolated CPE. Moreover, the presence of a carbapenemase gene did not necessarily correlate with phenotypic resistance. Two *E. coli* isolates carrying the complete *bla*_OXA-48_ gene were susceptible (as defined by breakpoints used for clinical isolates in human medicine) to ertapenem, imipenem, and meropenem. Similarly to our investigation, Ceccarelli et al. [9] screened samples from various food items and water in the Netherlands and found OXA carbapenemases in 0.16% of 4440 fecal samples from broilers, slaughter pigs, veal calves, and dairy cows. They demonstrated that the *bla*_OXA-like_ genes were chromosomally encoded and associated with *Shewanella* spp. rather than with CPE. This suggests that the partly high numbers of *bla*_OXA_-positive samples in our study were likely due to intrinsically resistant, environmental genera such as *Shewanella* rather than CPE, which implies a low risk for human health. This is supported by the fact that we detected *bla*_OXA-like_ genes on pig and broiler carcasses, but not in their cecum contents, which could indicate a contamination with environmental species during the slaughtering process.

We isolated CPE (*K. oxytoca*, *C. freundii,* and *E. coli* with *bla*_OXA-48_-like genes, *K. pneumoniae* with *bla*_KPC-3_ gene) only in the wastewater samples, while samples from slaughterhouses or meat products at retail were not affected. A recent study by Mueller et al. [15] found CPE significantly more often in clinical than in rural wastewater settings, suggesting a hospital-associated rather than a zoonotic origin. The lack of CPE in food items or along the sampled food chain demonstrates that previous reports indicating a low prevalence in German food-producing animals were still valid.

After the first identification of the *mcr-1* gene in 2015, numerous reports about its occurrence in humans in veterinary medicine demonstrated that the gene was globally disseminated, mainly in *E. coli* [11]. In this study, we found *mcr* genes in 18% of all 505 samples with the highest prevalence in samples from a poultry slaughterhouse (43%). In Germany, a recent study showed similar results. Borowiak et al. [16] detected *mcr* in 62.4% of 407 colistin-resistant *Salmonella* isolates during the period 2011–2018, which were associated with *mcr*-*1* (*n* = 175), *mcr*-*4* (*n* = 53), *mcr*-*5* (*n* = 18) or *mcr*-*1* and *mcr*-*9* (*n* = 8). When testing *E. coli* isolates from EU-monitoring programs (2011–2015), *mcr*-*1* was found in 3.8% of the samples, mostly in those from poultry farms (10%) [17]. As these isolates were collected without target screening for Col-E, this indicates a wide distribution. This is supported by a study demonstrating that *mcr*-*E. coli* were found in 26% of all pig farms in Germany [8].

The isolates carrying *mcr* in this study were mainly *E. coli* with a variety of different STs and serotypes indicating different sources rather than a common origin, although these findings are based on a limited number of ten isolates. In addition, we identified two *M. morganii* isolates carrying *mcr* genes, although this species is intrinsically colistin-resistant [18]. One explanation could be a co-selective effect of the *mcr* plasmid. However, no additional antibiotic resistances were detected for these isolates. Further studies are needed to elucidate the potential advantageous effects of carrying *mcr* plasmids despite intrinsic colistin resistance.

Surveillance of antimicrobial use on German farms showed a major decrease in the total amount of antimicrobial agents in the past five years (reduction by more than 50% between 2011 and 2017 [19]). This included a reduction of colistin use by 42%. Moreover, data from AST of commensal *E. coli* recently indicated that this decrease was followed by a reduction of antimicrobial resistance, particularly colistin resistance, in these bacteria [20]. There is a need to follow-up this tendency in the next years and to evaluate whether the prevalence of Col-E continues to decrease.

It remains an open question, how likely is it that the transmission of *mcr*-positive Enterobacterales to humans via the food chain occurs? Despite the high detection rates of *mcr* genes in water or livestock, a study consecutively testing 132 German travelers recently found that 11.4% of all persons carried *mcr*-*1*-positive Enterobacterales during and after travel, but none of them were colonized before leaving Germany [21]. In other European countries, the prevalence of *mcr*-*1* in the community or among patients in hospitals was also low (0–0.4%) [22,23,24]. The travel-associated acquisition of *mcr*-*E. coli* is in agreement with findings of *mcr*-*1* carriage rates of 5–88% in China, Bolivia, and Vietnam [25,26]. In these countries, reports indicate a livestock- or food-associated exchange of *mcr*-*E. coli*. For example, a study in Vietnam found that the colonization rate with *mcr-E. coli* was 9% in the urban population, but was 33% among farmers with exposure to poultry [27]. However, community-associated transmission as observed for ESBL-*E. coli* is also possible [28].

Historically, the detection of VRE in livestock became a major public health issue in the 1990s, when VRE was detected in food items and the occurrence of VRE was associated with the use of the glycopeptide avoparcin, which served as a growth promoter among food-producing animals [29]. Although in that time epidemiological studies partly suggested an association between VRE among humans and livestock, this correlation was never fully confirmed [30,31]. After the ban of avoparcin in veterinary medicine, VRE rates decreased, but many reports suggested that VRE persisted in livestock [32,33,34,35,36]. For Germany, one study showed the persistence of VRE among turkeys in 2009 [37], however, no current data on VRE prevalence in livestock are available. In this study, we detected no VRE in samples from the meat production chain, which indicates the absence or low prevalence in the observed region, though it might not be representative for Germany. VRE (*E. faecium*) was only isolated in environmental samples from municipal wastewater treatment plants. Genotyping showed that all isolates belonged to clonal complex (CC) 17, which is typically found in hospital-associated isolates [38]. This confirms the findings of a recent study, which systematically investigated the occurrence of several MDRB in wastewater [39]. Sources for these contaminations could be the community or hospital wastewater. In Germany, the number of VRE cases (associated with both *vanA* and *vanB*) in the healthcare setting has increased in the past years and hence, it is expected that an increasing number of colonized persons sheds VRE [13,40].

In addition to CPE, Col-E and VRE, we also cultured ESBL- and AmpC-producing Enterobacterales and found them in 12–46% of all samples. The results were expected and confirmed those of previous studies demonstrating the widespread occurrence of these bacteria among German food-producing animals.

In summary, the overall prevalence of MDRB within the meat production chain was medium to low for the analyzed groups (ESBL, Col-E, CPE, VRE). In comparison, the risk of transmission via the consumption of contaminated food remains the highest for ESBL/AmpC-producers. Though relatively high numbers of the *bla*_OXA-like_ genes were found, the number of isolated CPE was low. VRE detected in the urban environment are more likely to have a human rather than a zoonotic origin. However, this study is limited by its relatively small sampled area, which implies reduced representativeness. Furthermore, a combined approach was only applied for Col-E and CPE, but not for VRE and ESBL. This was because culture-based protocols for these bacteria are well established. However, it cannot be ruled out that more positive samples would have been found in a PCR for these groups as well. We conclude that despite the growing prevalence of MDRB being one of the major global threats for public health, the risk of zoonotic transmission via the food chain within the monitored area is moderate. To further improve monitoring programs, more sensitive testing strategies to facilitate the distinction of potentially pathogenic CPE and intrinsically carbapenem-resistant environmental species need to be developed. We have shown that a combination of PCR and culture-based techniques could be a useful basis for this.

## 4. Materials and Methods

### 4.1. Sample Collection

In total, 505 samples were analyzed (see Figure 5 for details). In the course of the EU-wide zoonoses-monitoring, samples were collected from two slaughterhouses (one for pigs, one for broilers) and from fresh food products at supermarkets. The samples from the EU monitoring were complemented with samples from different meat-processing plants and from environmental sources to cover all the stages of the meat production chain. All samples were taken in seven different districts of East Westphalia-Lippe, a region in northwestern Germany with a population of about two million citizens.

Freshly opened cecum contents were analyzed to assess the MDRB carriage status of the animals arriving in the slaughterhouses, i.e., potential MDRB that were acquired during fattening on the farms. Samples from carcasses were collected at the end of the slaughtering procedure to test for potential contaminations during the process. Therefore, pig carcasses were swabbed with sampling sponges and neck skin-samples were taken from broilers. At meat-processing plants, samples of meat juices were taken from the transport boxes and drains were swabbed to test for MDRB that might have been released into the environment via wastewater. A variety of different food products from supermarkets was analyzed to test for contaminations that consumers were directly exposed to. Finally, samples from water, soil, and municipal wastewater treatment plants were collected to assess the prevalence of MDRB in the environment.

### 4.2. Molecular and Culture-Based Screening for Col-E, CPE, VRE, and ESBL-Producers

For all sample types, a selective enrichment from the native material was incubated overnight at 37 °C. Enrichment media were composed of buffered peptone water (BPW) (Merck, Darmstadt, Germany) with 4 mg/L colistin for Col-E detection and 0.25 mg/L ertapenem plus 50 mg/L vancomycin for CPE detection. Up to five overnight enrichments were pooled and DNA was extracted using the DNeasy Blood & Tissue Kit (Qiagen, Hilden, Germany). DNA pools were screened for the presence of *mcr1–mcr5* in a multiplex PCR following the protocol by Rebelo et al. [41] and for CPE using the Check-Direct CPE Kit (Check-Points B.V., Wageningen, The Netherlands). For positive pools, the procedure was repeated on individual samples. From positive samples, isolates were cultured on ChromAgar COL-APSE (Mast Group Ltd., Bootle, UK), MacConkey agar plates supplemented with 2 mg/L colistin, and MacConkey agar plates with 4 mg/L colistin for Col-E and chromID CARBA SMART agar (bioMérieux, Marcy l’ Etoile, France) for CPE. ESBL/AmpC producing Enterobacterales were detected following the EURL-AR recommendations [42]. For the cultural detection of VRE, native material was incubated in ChromoCult enterococci broth (Merck, Darmstadt, Germany) at 37 °C overnight. The enrichment was plated on chromID VRE selective agar (bioMérieux, Marcy l’ Etoile, France) and incubated at 37 °C for 24–48 h.

Species of sub-cultured isolates were identified using MALDI-TOF (Bruker Daltonik, Bremen, Germany). Minimum inhibitory concentrations (MICs) were determined using broth microdilution (MICRONAUT-S, MERLIN Diagnostika Gmbh, Bornheim, Germany) with the exception of ertapenem, imipenem, and meropenem, where E-tests were used (bioMérieux, Marcy l’ Etoile, France).

### 4.3. Molecular Typing of Selected Isolates

Twenty-three isolates were selected for molecular typing using whole genome sequencing (WGS). These isolates included all isolated CPE (*n* = 8), vancomycin-resistant *E. faecium* (*n* = 4) and Col-E carrying *mcr-5* (*n* = 1). From all the *mcr-1*-positive isolates, ten were randomly selected for sequencing. Genomic DNA (gDNA) was extracted using the NEB Monarch Genomic Purification Kit (New England Biolabs, Ipswich, MA, USA). Isolates were sequenced on a PacBio Sequel II system (Pacific Biosciences, Menlo Park, CA, USA) using a 20 kb insert size library and the SMRTbell^®^ Express Template Prep Kit 2.0. Raw sequences were assembled de novo using the SMRTLink software suite v8 with default parameters for a microbial assembly. The resulting sequences are available at Genbank under BioProject number PRJNA662907. From the resulting contig sequences, multilocus sequence types (MLST), in silico serotypes and core genome MLST (cgMLST) profiles were created using the Ridom SeqSphere^+^ software version 4 (Ridom GmbH, Münster, Germany). All sequences were screened for the presence of resistance genes using ABRicate v0.9.8 (https://github.com/tseemann/abricate) and the ResFinder database [43]. Replicon family types of plasmids carrying the *mcr* gene were identified using PlasmidFinder [44]. The sequences of the contigs containing the *mcr* gene were compared with the nucleotide collection of the National Center for Biotechnology Information (NCBI) (https://blast.ncbi.nlm.nih.gov/Blast.cgi?PAGE_TYPE=BlastSearch) using the BLAST algorithm [45].

## Figures and Tables

**Figure 1 antibiotics-09-00708-f001:**
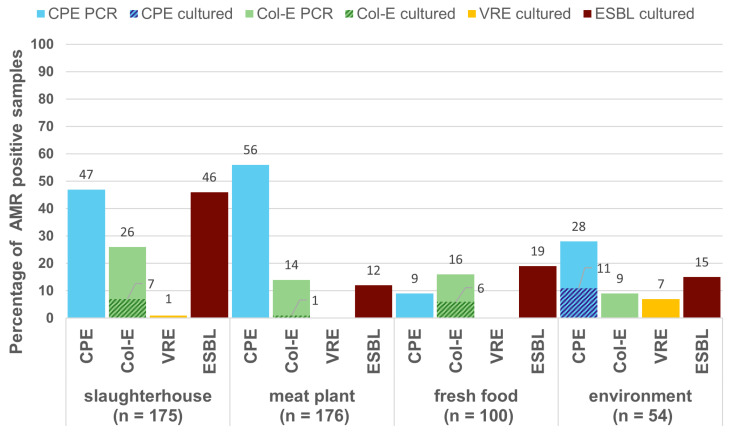
Overview of carbapenemase-producing Enterobacterales (CPE), colistin-resistant Enterobacterales (Col-E), vancomycin-resistant enterococci (VRE), and extended-spectrum β-lactamase (ESBL)-producing *Escherichia coli* prevalence in different stages of the meat production chain. For CPE and Col-E, stippled areas indicate from how many samples’ isolates were successfully cultured in comparison to the relative amount of samples that were positive for antimicrobial resistance (AMR) genes in the PCR screening.

**Figure 2 antibiotics-09-00708-f002:**
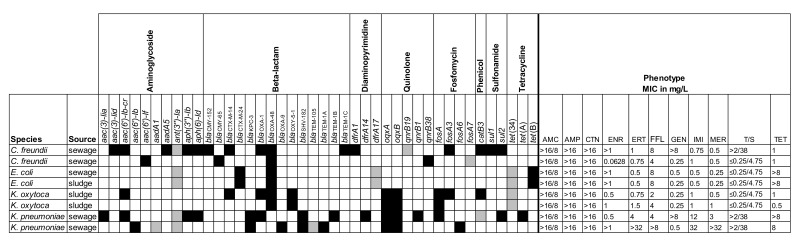
Resistance profiles for eight carbapenemase-producing Enterobacterales. Overview of the presence/absence of the resistance genes and minimum inhibitory concentrations (MICs) for the selected antimicrobial agents. AMC = amoxicillin/clavulanic acid; AMP = ampicillin; CTN = cephalothin; ENR = enrofloxacin; ERT = ertapenem; FFL = florfenicol; GEN = gentamicin; IMI = imipenem; MER = meropenem; T/S = trimethoprim/sulfamethoxazole; TET = tetracycline. Black = presence (>80% coverage, 95% identity); grey = partial presence (<80% coverage); white = absence.

**Figure 3 antibiotics-09-00708-f003:**
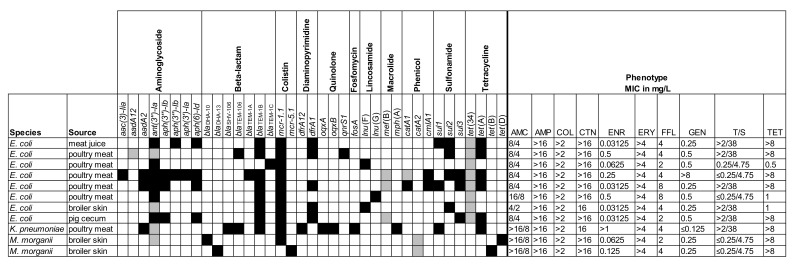
Resistance profiles for eleven colistin-resistant Enterobacterales. Overview of the presence/absence of the resistance marker genes and minimum inhibitory concentrations (MICs) for the selected antimicrobial agents. AMC = amoxicillin/clavulanic acid; AMP = ampicillin; COL = colistin; CTN = cephalothin; ENR = enrofloxacin; ERY = erythromycin; FFL = florfenicol; GEN = gentamicin; T/S = trimethoprim/sulfamethoxazole; TET = tetracycline. Black = presence (>80% coverage, 95% identity); grey = partial presence (<80% coverage); white = absence.

**Figure 4 antibiotics-09-00708-f004:**
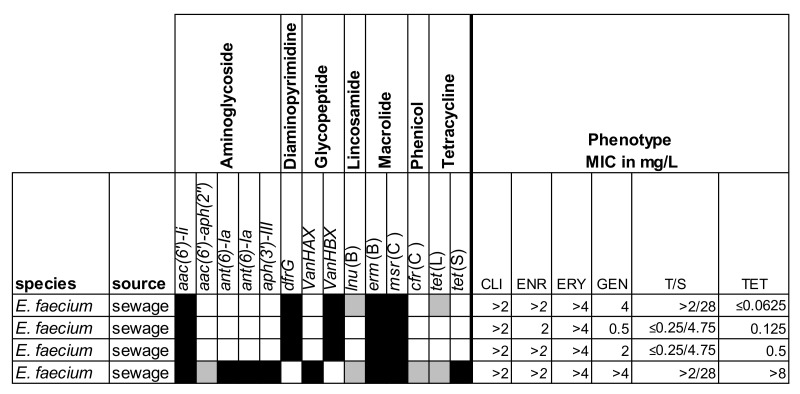
Resistance profiles for four vancomycin-resistant enterococci. Overview of the presence/absence of the resistance marker genes and minimum inhibitory concentrations (MICs) for the selected antibiotics. CLI = clindamycin; CMP = chloramphenicol; ENR = enrofloxacin; ERY = erythromycin; GEN = gentamicin; T/S = trimethoprim/sulfamethoxazole; TET = tetracycline. Black = presence (>80% coverage, 95% identity); grey = partial presence (<80% coverage); white = absence.

**Figure 5 antibiotics-09-00708-f005:**
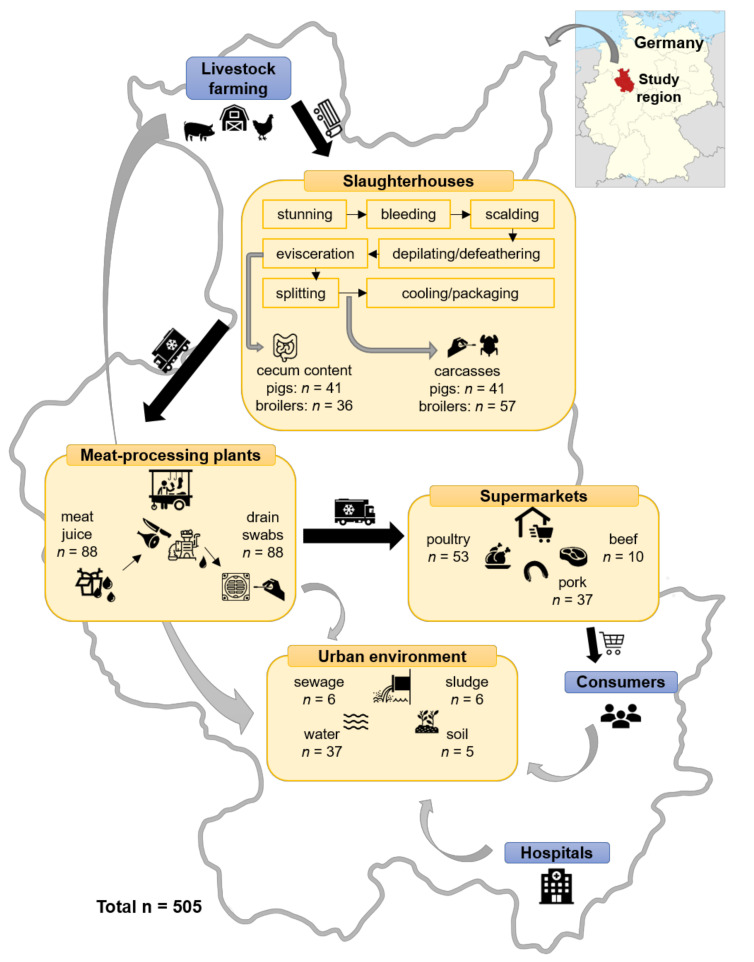
Overview of samples taken at different stages of the meat production chain. Locator map of the study region by TUBS, available from https://commons.wikimedia.org/wiki/File:Locator_map_RB_DT_in_Germany.svg, is licensed under CC BY-SA 3.0.

**Table 1 antibiotics-09-00708-t001:** Number of positive samples in a direct PCR and the number of cultured isolates for carbapenemase-producing Enterobacterales (CPE), colistin-resistant Enterobacterales (Col-E), vancomycin-resistant enterococci (VRE), and extended-spectrum β-lactamase (ESBL)-producing *Escherichia coli* in different specimens within the meat production chain.

Specimen	No. of Samples	No. of Positive Samples in Direct PCR		No. of Isolates Cultured from Samples			
		Carbapene-Mases (*n*)	*mcr* (*n*)	CPE	Col-E	VRE	ESBL
Carcasses broilers	57	*bla*_OXA-48_ (57)	*mcr-1* (20), *mcr-5* (22) **	0	*mcr-1* (6), *mcr-5* (1)	1	8 *
Carcasses pigs	41	*bla*_OXA-48_ (25)	*mcr-2* (1), *mcr-4* (2)	0	0	0	n.a.
Cecum broilers	36	–	*mcr-1* (6), *mcr-5* (1)	0	*mcr-1* (5)	0	20
Cecum pigs	41	–	*mcr-1* (3)	0	*mcr-1* (1)	0	17
Drain swab	88	*bla*_OXA-48_ (60)	*mcr-1* (1), *mcr-4* (14)	0	*mcr-1* (1)	0	8
Meat juice	88	*bla*_OXA-48_ (38)	*mcr-1* (4), *mcr-4* (5)	0	*mcr-1*(1)	0	13
Beef	10	*bla*_OXA-48_ (1)	-	0	0	0	1
Pork	37	*bla*_OXA-48_ (4)	*mcr-1* (1), *mcr-4* (1), *mcr-5* (1)	0	0	0	3
Poultry	53	*bla*_OXA-48_ (4)	*mcr-1* (13)	0	*mcr-1* (6)	0	15
Water	37	bla_OXA-48_ (6)	*mcr-4* (1)	0	0	0	0
Sewage/Sludge/Soil	17	*bla*_OXA-48_ (7), *bla*_KPC_ (2), *bla*_VIM/NDM_ *(*3)	*mcr-1* (4)	*bla*_OXA-48_ (6), *bla*_KPC_ (2)	0	4	8

* only 21 of 57 samples were tested; ** 9 samples were positive for *mcr-1* and *mcr-5.*

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
