# Peer review of "Prevalence and Epidemiology of Multidrug-Resistant Pathogens in the Food Chain and the Urban Environment in Northwestern Germany"

_antibiotics, 2020, doi:10.3390/antibiotics9100708_

Round 1

Reviewer 1 Report

The manuscript is well written, with methods appropriately used.  It concludes that there is low risk of MDRB spreading through meat consumption/production food chain.

Following suggestions should be taken into consideration to make the manuscript stronger

It appears that PCR based methods detected colistin (mcr) and carbapenemase (blaOXA-48, blaKPC, blaNDM, blaVIM) resistance genes when culture based methods were unsuccessful in their detection.  For vancomycin resistance, however, only culture based method was conducted and, therefore, it cannot be confirmed that they were absent in 3 of the 4 types of samples taken from 4 different stages of meat processing, respectively.

Suggest using pooled reactions to test for vanA and vanB by PCR or justify why this was not tested. The conclusion that VRE detected in the urban environment are more likely to have a human rather than a zoonotic origin also needs to be modified accordingly, as also lines 216-217.

Grammatic usage in terms of tenses used is not consistent across the whole text. These should be corrected. I have attempted to do a few.

Minor suggestions:

Please define MDRB. (resistance to 3 or more classes of drugs?)

Abstract

  1. Line 19 Change ‘i.e’. to viz., (namely)
  2. Line 20 Move (Col-E) to the end after Enterobacterales, in “colistin-resistant (Col-E) Enterobacterales”
  3. Line 21 change therefore to altogether

Introduction

Line 59: Put period after environment and start new sentence “Additional genotypic and phenotypic investigations were performed as follows:

Line 69: Change shall to should “Therefore, this combined analysis should  improve CPE detection.”

Line 72 Change ‘was’ to ‘is’

Result

Figure 1: Change Y axis legend to “Percentage of AMR positive samples”

Lines 94-95 :  18% of how many total samples .  Is it 505? OR is it 23 isolates?  Please put n= 

Discussion

Line 201: Change “indicating” to indicate

Reviewer 2 Report

Kless et al in the paper titled “Prevalence and epidemiology of multidrug-resistant pathogens in the food chain and the urban environment in northwestern Germany” describes the presence of multidrug-resistant bacteria in a region in Northwestern Germany. The authors also tried to increase the sensitivity of traditional culture method with molecular and culture-based approach. There are some issues with the presentation of the results.

  1. In the introduction section, it is not clear why the authors decided to combine molecular technique with the traditional technique. What does the author mean with molecular technique? Is it PCR or genotyping or both? Are there any previous literature mentioning the limitation of culture-only technique that warrants combination with molecular technique? Is the combination never previously applied in the region?
  2. There are no results which state that combination with molecular technique increases the sensitivity of detection although this seems to be one of the purposes of the study. Again, it is not clear with molecular technique does the author refer to. If it is the PCR, the authors should make it clear that PCR detects more antimicrobial resistance genes. To be fair, this can be understood from Figure 1, but because PCR is a very common molecular technique for detection of AMR genes, as a reader, we would not expect that PCR is the molecular technique introduced in this paper. Therefore, instead of the phrase ‘molecular technique’ it is better to use a clearer description such as PCR and sequencing.
  3. If the presence of antimicrobial genes does not always mean resistance phenotype, must genotyping always be performed. Please discuss this further and what steps should be taken to reduce false positive (such PCR of few body sites or places).
  4. For clarity of Figure 1 and Table 1, it is best that Figure 5 is discussed first in the beginning of the paper.
  5. In both Figure 1 and Table 1, VRE and ESBL are not detected with PCR. Please discuss why.
  6. Please check again the numbers on Figure 1 and Table 1 because PCR percentage of mcr in slaughterhouses samples do not match. 31% in table 1 and 26% in Figure 1.
  7. In the discussion line 228, the authors suggest that “In summary, the overall prevalence of MDRB within the meat production chain was low for analyzed groups”, although there is an up to 47% and 46% prevalence of CPE and ESBL, respectively in slaughterhouse and up to 56% prevalence of CPE in meat plant. I think, this prevalence is not low.

Reviewer 3 Report

This is a good paper providing a knowledge on multidrug-resistant pathogens in food and environment in northwestern Germany. I suggest that some improvements should be made.

  1. Line 57: Is this EU-wide monitoring program of 2018/2019 concordance with Directive and/or Decision of EU? Please provide the information on Directive and/or Decision of EU.

  1. Line 60-73: These texts should be shortened here and/or moved to “Study design” of Material and Methods section

  1. Although the authors have speculated that mcr-1-containing isolates have large heterogeneity, the authors randomly selected isolates and sequenced a limited number (10) of samples by Whole genome sequencing (WGS). Analysis of more than 15 samples were required, and then these data of large heterogeneity should be confirmed by PFGE.

  1. WGS data should be deposited. Then, accession numbers such as European Nucleotide Archives accession number, or GenBank Accession number should be shown.

  1. Line 300-302: Accession number of the reference plasmids should be shown here or in Tables.

Round 2

Reviewer 2 Report

The authors have addressed my concerns sufficiently. Therefore, I would like to recommend this manuscript to be published as it is.

Reviewer 3 Report

The manuscript has been significantly improved by the revision.